# Mediating Effects of Emotional Symptoms on the Association between Homophobic Bullying Victimization and Problematic Internet/Smartphone Use among Gay and Bisexual Men in Taiwan

**DOI:** 10.3390/ijerph17103386

**Published:** 2020-05-13

**Authors:** Dian-Jeng Li, Yu-Ping Chang, Yi-Lung Chen, Cheng-Fang Yen

**Affiliations:** 1Graduate Institute of Medicine, College of Medicine, Kaohsiung Medical University, Kaohsiung 80708, Taiwan; u108800004@kmu.edu.tw; 2Department of Addiction Science, Kaohsiung Municipal Kai-Syuan Psychiatric Hospital, Kaohsiung 80276, Taiwan; 3School of Nursing, The State University of New York, University at Buffalo, Buffalo, NY 14214-3079, USA; yc73@buffalo.edu; 4Department of Healthcare Administration, Asia University, Taichung 41354, Taiwan; 5Department of Psychology, Asia University, Taichung 41354, Taiwan; 6Department of Psychiatry, Kaohsiung Medical University Hospital, Kaohsiung 80708, Taiwan

**Keywords:** sexual minority, problematic internet use, smartphone, emotional symptom, homophobic bullying

## Abstract

Problematic internet/smartphone use (PI/SU) and homophobic bullying has become a new type of mental health problem among sexual minorities. However, few studies have investigated the mediators of the association between these factors. We aimed to develop a model to estimate the mediating effect of emotional symptoms, including depression and anxiety, on this association among gay and bisexual men in Taiwan. In total, 500 gay or bisexual men in early adulthood were recruited, and their histories of homophobic bullying victimization during childhood and adolescence, current severity of PI/SU, and current emotional symptoms were evaluated using self-administered questionnaires. A mediation model was developed to test the mediating effect of emotional symptoms on the association between homophobic bullying victimization and PI/SU. In total, 190 (38%) and 201 (40.2%) of the participants had experiences of traditional and cyber homophobic bullying victimization, respectively. A higher level of homophobic bullying victimization was significantly associated with a more severe level of PI/SU, and this was mediated by a higher severity of emotional symptoms. There was a significant effect of emotional symptoms on the association between homophobic bullying victimization and PI/SU. Timely interventions for emotional symptoms are necessary for gay and bisexual men, especially for those who are victims of homophobic bullying.

## 1. Introduction

### 1.1. Problematic Internet/Smartphone Use Among Sexual Minorities

In recent decades, browsing the internet has become increasingly popular among children and adolescents. Digital games, despite being potentially beneficial in improving short-term memory [1], have been associated with a detrimental effect on mental health, similar to problematic substance use [2,3]. Smartphone use has also increased, and it is associated with enhancing productivity, information seeking, social communication, relaxation, and entertainment [4]. However, problematic internet/smartphone use (PI/SU) has become a new type of mental health problem [5] associated with significant negative consequences. For example, problematic internet use has been associated with anxiety, depression, psychotic symptoms, alcohol abuse, hostility, and even suicidality [5,6]. Similarly, problematic smartphone use has also been associated with mental health problems, such as depression, anxiety, and sleep problems [7], and physical problems, such as blurred vision, neck stiffness, and somatic pain [6,7]. Heavy mobile phone use can hinder an individual’s social-economic function by reducing school performance and interpersonal social interactions [5,8]. However, very few studies on the relatively new field of non-substance addictions have discussed sexual minorities. An online survey indicated that problematic gaming and internet use were significantly more prevalent in sexual minorities than in heterosexuals [9], although another study recruiting transgender subjects did not demonstrate an overrepresentation of internet gaming disorder [10]. Meyer proposed the minority stress theory to address the association between multiple stressors and mental health status in sexual minorities [11]. This model was later applied to substance use in sexual minorities, and showed an adequate fit for this population [12]. However, further studies are warranted to better understand the generalization of this model to behavioral addictions.

### 1.2. Effects of Homophobic Bullying on Mental Health

Homophobic bullying is common in sexual minorities, causing substantial negative impacts on their mental health. A previous meta-analysis demonstrated that sexual minority youths reported higher rates of violence and victimization in comparison with heterosexual youths [13]. Rates of suicidal ideation have also been reported to be higher among sexual minority youths with a history of bullying victimization [14]. Homophobic bullying in childhood and adolescence results in harmful effects on future psychosocial and health outcomes in adulthood [15]. It may interfere with developing positive self-concepts [16] and contribute to the development of mental disorders among sexual minorities [17]. On the other hand, minority stress theory also emphasizes that disparities between sexual minorities and heterosexuals can be attributed to stigma, discrimination, and victimization experiences as a result of a homophobic environment [11,18].

### 1.3. Aims of the Current Study

According to the Taiwan Social Change Survey 2012, 4.4% of Taiwan’s population identified themselves as non-heterosexual [19]. The health and well-being of sexual minorities had been significantly improved in Taiwan, and it even became the pioneer in Asia [19]. The polytheism in Taiwanese folk beliefs, the tolerance to different religions, and the diversified culture may contribute to the relatively friendly society in Taiwan. However, sexual minorities in Taiwan still suffered from discrimination and homophobic bullying, leading to suicidality [20] and poor quality of life [21]. The impact of homophobic bullying and PI/SU has been identified in sexual minorities, but few studies have explored the association between them. The relationship between the experience of homophobic bullying victimization in childhood and adolescence and PI/SU has been verified in sexual minority youths [22]; however, studies exploring the detailed etiology of this association are lacking. Early identification of factors influencing the association between homophobic bullying and PI/SU will help clinicians to provide timely interventions.

As mentioned above, PI/SU has been associated with anxiety and depression within the general population; therefore, we hypothesized that emotional symptoms may play a mediating role in PI/SU among sexual minorities who experienced homophobic bullying victimization during their childhood and adolescence. Accordingly, the aim of this study was to investigate the mediating effect of emotional symptoms, including depression and anxiety, on the association between homophobic bullying victimization and later PI/SU.

## 2. Methods

### 2.1. Participants

We recruited men who identified as being gay or bisexual aged from 20 to 25 years old. In the current study, young adults were targeted to reduce the possibility of recall bias for their experience of homophobic bullying during childhood and adolescents. Individuals showing any cognitive decline that may have hindered them from understanding the purpose of the study or completing the questionnaires were excluded at the initial screening stage.

### 2.2. Procedure

The detailed procedure of the current study was described in our previous study [22]. In brief, we recruited participants through online and printed advertisements posted on lesbian, gay, bisexual, and transgender (LGBT) clubs, along with social networking sites, such as Facebook, bulletin board system, and the home pages of five health promotion and counseling centers for sexual minorities in Taiwan. The included participants completed the questionnaires after signing informed consent, and there was no follow-up period as a cross-sectional study. Research assistants explained the procedure to help the participants complete the questionnaires. The assessment of special issues in the study were also explained, including current status of internet/smartphone use, experience of homophobic bullying, experience of cyberbullying, and current level of depression along with anxiety. This study was approved by the Institutional Review Board of Kaohsiung Medical University Hospital (KMUHIRB-F(I)-20150026).

### 2.3. Measures

#### 2.3.1. Problematic Internet and Smartphone Use

The self-reported Chen Internet Addiction Scale (CIAS) was used to estimate the severity of problematic internet use in the month preceding the study. The CIAS is a 26-item questionnaire rated on a 4-point Likert scale as follows: 0 = not at all, 1 = disagree, 2 = agree, and 3 = entirely agree. The total scores ranged from 26 to 104 [23], with a higher total CIAS score representing a more severe level of problematic internet use. The internal consistency of CIAS was around 0.79 to 0.93, and the test-retest reliability (internal validity) was 0.83 [23]. In this study, the internal reliability (Cronbach’s α) was 0.93.

The Smartphone Addiction Inventory (SPAI) was used to evaluate the participants’ severity of problematic smartphone use in the month preceding the study, as a self-administered questionnaire. It is also a 26-item questionnaire rated on a 4-point Likert scale as follows: 0 = never, 1 = just a little, 2 = often, and 3 = all the time. The total scores ranged from 26 to 104 [24], with a higher total score indicating a more severe level of problematic smartphone use. The exploratory factor analysis for SPAI demonstrated Kaiser–Meyer–Olkin (KMO) value at 0.93 and significance (*p* < 0.001) of the Bartlett test, indicating good construct validity. The Cronbach’s α of the SPAI in the present study was 0.93.

#### 2.3.2. Homophobic Traditional and Cyberbullying Victimization

The 6-item Chinese version of the self-administered School Bullying Experience Questionnaire (C-SBEQ) [25] was used to estimate the experiences of homophobic traditional bullying victimization during childhood and adolescence, including social exclusion, name calling, verbal abuse, physical abuse, forced work, and confiscation of money, school supplies and snacks at school, tutoring schools, after-school classes, and part-time workplaces. The traditional bullying was defined as forms of bullying prior to digital media. We encouraged the participants to recall these experiences from when they attended elementary and high school. The experiences of traditional homophobic bullying victimization due to gender role nonconformity (6 items) and disclosure of sexual orientation (6 items) were evaluated separately. Each item was graded on a 4-point Likert scale as follows: 0 = never, 1 = just a little, 2 = often, and 3 = all the time. The total score ranged from 0 to 36, with a higher total C-SBEQ score indicating more severe victimization of homophobic traditional bullying. The C-SBEQ has been reported to have good reliability and validity [25]. The Cronbach’s α values of the scales for measuring the two types of bullying victimization due to gender nonconformity and sexual orientation were 0.79 and 0.82, respectively. Those who answered 2 or 3 on any item were classified as being victims of homophobic traditional bullying.

In order to evaluate the experience of cyberbullying victimization during childhood and adolescence, three items from the Cyberbullying Experiences Questionnaire (CEQ) [26] were used with multiple categories of assessments, including the experience of others posting mean or unpleasant comments, others posting upsetting pictures, photos, or videos, and online rumor-spreading through emails, blogs, social media platforms, and pictures or videos during the aforementioned school stages. We assessed the experiences of homophobic cyberbullying victimization due to gender role nonconformity (3 items) and disclosure of sexual orientation (3 items) separately. Each item was graded on a 4-point Likert scale as with the C-SBEQ, with the total score ranging from 0 to 18 and a higher total score indicating more severe victimization of homophobic cyberbullying. The Cronbach’s α values of the scales for measuring cyberbullying victimization due to gender nonconformity and sexual orientation were 0.71 and 0.86, respectively. Those who answered 1 or higher on any item were classified as being victims of homophobic cyberbullying.

#### 2.3.3. Emotional Symptoms

We estimated the severity of emotional symptoms according to the two following domains: depressive and anxiety symptoms. The Mandarin Chinese version of the Center for Epidemiological Studies-Depression Scale (MC-CES-D) was used to assess the frequency of depressive symptoms in the week preceding the study [27,28]. There were 20 items and each item was scored on a 4-point scale as follows: 0 = never or less than one day per week, 1 = one to two days per week, 2 = three to four days per week, and 3 = five to seven days per week. The total score ranged from 0 to 60, with a higher total score representing a more severe level of depression.

The test retest reliability (internal validity) of CES-D were in the moderate range from 0.45 to 0.70. Cronbach’s alpha for the MC-CES-D in this study was 0.92.

To evaluate the severity of anxiety, the 20-item self-administered State-Trait Anxiety Inventory (STAI-S) form Y was used [29,30]. Every item was graded on a 4-point Likert scale as follows: 1 = never, 2 = just a little, 3 = often, and 4 = all the time. The total score ranged from 20 to 80, with a higher total score indicating a more severe level of anxiety. The construct validity of the STAI demonstrated the KMO value at 0.824 and significance (*p* < 0.001) of the Bartlett test, indicating good construct validity. Cronbach’s alpha for the STAI-S in the present study was 0.87.

### 2.4. Statistical Analysis

The hypothesized model for the association between homophobic bullying, emotional symptoms, and PI/SU is illustrated in Figure 1. SPSS and AMOS version 23.0 for Windows (SPSS Inc., Chicago, IL, USA) were used to conduct the analysis. We examined bivariate associations using Pearson correlation analysis. Two-step structural equation modeling was used in the current study. Initially, confirmatory factor analysis was used to verify the relationships between latent variables and their indicators in the measurement model. Latent variable path analysis with maximum likelihood parameter estimation was then used to estimate the model adequacy and the direct/indirect effects of homophobic bullying on PI/SU through emotional symptoms [31]. Standardized estimates (beta coefficient; β) were reported for the predictive strength explained in the model.

The Sobel test was used to examine the mediating effect of emotional symptoms on the relationship between homophobic bullying and PI/SU [32]. Furthermore, in order to test the adequacy of the model, multiple indices were used to verify the goodness of model fit. For each of these fit indices, the values indicating acceptable model fit were as follows: chi-square goodness-of-fit test (χ^2^/df < 5.0); non-significant χ^2^ (*p* > 0.05); Comparative Fit Index (CFI ≥ 0.95); Incremental Fit Index (IFI ≥ 0.95); Tucker–Lewis Index (TLI ≥ 0.95); Incremental Fit Index (IFI ≥ 0.95); Goodness of Fit Index (GFI ≥ 0.95); Adjusted Goodness of Fit Index (AGFI ≥ 0.95); Normed Fit Index (NFI ≥ 0.95); Root-mean Square Error of Approximation (RMSEA < 0.08); and Standardized Root Mean Square Residual (SRMR ≤ 0.05) [31,33,34].

## 3. Results

### 3.1. Descriptive Statistics and Correlation Matrix

A total of 500 males (129 bisexual men and 371 gay men) recruited in this study with a mean age of 22.94 ± 1.6 years. Regarding the education level, 450 (90%) of the subjects graduated from college or a graduate institute, and the others (10%) were from high school or below. The means and standard deviations of variables regarding levels of homophobic bullying victimization, PI/SU, symptoms of anxiety, and depression are listed in Table 1. The correlation matrix with significance is also demonstrated. In total, 190 (38%) and 201 (40.2%) of the participants had experiences of traditional and cyber homophobic bullying victimization during childhood and adolescence, respectively. Correlations of all of the measured variables remained significant.

### 3.2. Tests for the Mediation Model and Estimated Co-Efficient of Paths

The structural equation models estimating the direct and indirect effects and the estimated co-efficient of paths in the hypothesized model are presented in Figure 2. We found statistical significance for an indirect effect of 0.148 (the Sobel test was Z = 4.74; *p* < 0.05) based on the product terms of the path from homophobic bullying victimization to emotional symptoms (β = 0.36, *p* < 0.001) and path from emotional symptoms to PI/SU (β = 0.41, *p* < 0.001). However, the direct effect showed no statistical significance. These results confirmed the mediating effect of emotional symptoms on the association between homophobic bullying victimization and PI/SU. Furthermore, multiple goodness-of-fit indices showed that the hypothesized model had an excellent fit for χ^2^/df, *p* value of χ^2^, RMSEA, GFI, AGFI, NFI, CFI, IFI, TLI, and SRMR, indicating that our hypothesized mediation model was a good fit (Table 2).

## 4. Discussion

In the current study, we found that a higher level of homophobic bullying victimization during childhood and adolescence was significantly associated with a higher level of PI/SU during early adulthood, and this this was mediated by emotional symptoms. The goodness-of-fit of multiple indices demonstrated the adequacy and applicability of the conceptual model.

### 4.1. Association from Homophobic Bullying Victimization to PI/SU

Research in heterosexual individuals has demonstrated significant relationships between bullying victimization and problematic internet use. For example, one cross-sectional study revealed that both traditional bullying and cyberbullying victimization had significant impacts on problematic internet use in youths [35]. For sexual minorities, we previously reported that homophobic bullying victimization was associated with PI/SU [22] and substance use [36,37] among gay and bisexual men. Although there are some differences between substance addition and non-substance addiction, they may share some etiological similarities. Regarding the relationship between homophobic bullying victimization and PI/SU, the minority stress hypothesis [11,18], psychological mediation hypothesis [16], and ecological system hypothesis [38] may reveal the fundamentals for a better understanding of the etiology. On the other hand, the internet may serve as an anonymous environment where sexual minorities can feel at ease. A previous study demonstrated that sexual minority youths might feel supported on the internet and make more online friends than heterosexual youths [39]. Another study also indicated that sexual minority youths felt that social interactions on the internet were more comfortable than in-person relationships [40]. We propose that such characteristics may play an important role in developing PI/SU within sexual minorities.

### 4.2. The Mediating Effect of Emotional Symptoms

The negative effects of homophobic bullying victimization on mental health have been investigated within sexual minorities, and associations between homophonic bullying victimization and depression/anxiety have been reported [41]. A systemic review also revealed that bullying victimization is predominantly related to depression within sexual minority youths [42]. The relationship between emotional symptoms and PI/SU has also been explored. Low emotional stability and depression have been associated with problematic mobile phone use [43], and Lee reported that heavy internet and smartphone use could be predicted by social interaction anxiety [44]. Moreover, more severe levels of depression and anxiety have been correlated with a higher total Smartphone Addiction Scale score [45]. A conceptual model suggested that reassurance-seeking behavior could explain this association, where the negative reinforcement of PI/SU may further exaggerate depression and anxiety [46]. We further confirmed the mediating effect of emotional symptoms on the association from homophobic bullying victimization to PI/SU. PI/SU has been reported to have a reinforcing effect that increases the craving for further use to relieve anxiety and depression [7], and this may represent PI/SU within sexual minorities suffering from homophobic bullying victimization.

### 4.3. Limitations

In addition to the connection from homophobic bullying victimization to PI/SU, the current study further explored the mediating effect of emotional symptoms, which has rarely been reported previously. However, several limitations need to be discussed. First, the self-reported questionnaires of PI/SU are insufficient to make a clinical diagnosis. Second, as the depression and anxiety scales were also self-administered, the present study could not determine diagnoses of depressive/anxiety disorders or other psychiatric comorbidities. Third, the self-administered questionnaires retrospectively acquired information on homophobic bullying victimization, so there may have been recall bias. Finally, this study only recruited gay and bisexual men, so the findings may not be generalizability to other populations, such as lesbian or other sexual minorities.

## 5. Conclusions

The current study found that a higher degree of homophobic bullying victimization was significantly associated with a higher level of PI/SU among gay and bisexual men, and that this was mediated by emotional symptoms. In order to prevent the onset of PI/SU, we suggest that mental health professionals should make early interventions for emotional symptoms for gay and bisexual men with previous experiences of homophobic bullying victimization.

## Figures and Tables

**Figure 1 ijerph-17-03386-f001:**
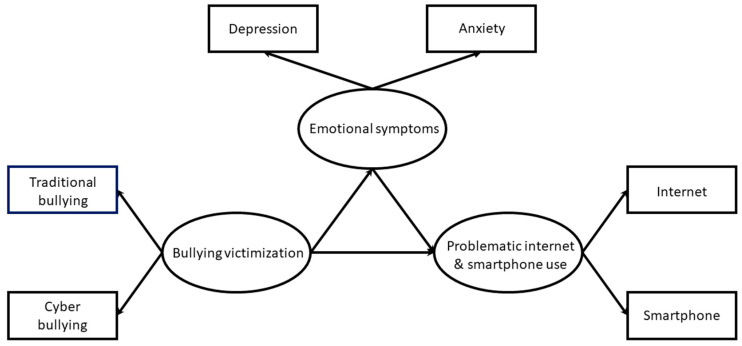
The conceptual model of mediating effect.

**Figure 2 ijerph-17-03386-f002:**
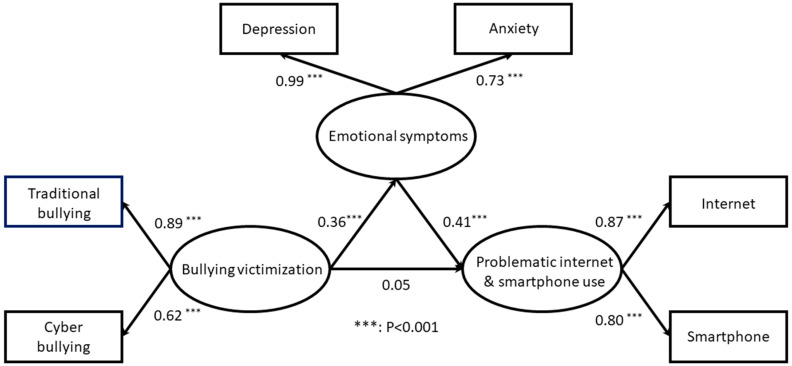
Final model of mediating effect indicating the estimated coefficients of the paths and factorial loadings.

**Table 1 ijerph-17-03386-t001:** The correlation matrix of observation variables.

Variables	Mean	SD	1	2	3	4	5	6
1. Depression	17.47	10.32	-	0.73 *	0.38 *	0.34 *	0.32 *	0.21 *
2. Anxiety	20.40	11.60		-	0.26 *	0.26 *	0.24 *	0.20 *
3. Problematic internet use	58.29	14.18			-	0.70 *	0.13 *	0.13 *
4. Problematic smartphone use	62.16	14.18				-	0.16 *	0.19 *
5. Traditional bullying victimization	5.01	5.03					-	0.55 *
6. Cyberbullying victimization	1.28	2.29						-

* *p* < 0.001

**Table 2 ijerph-17-03386-t002:** The indices of goodness-of-fit index for the mediation model.

Goodness of Fit Index	Estimates	Acceptable Ranges
χ^2^/df	1.999	<5.0
P value of χ^2^	0.06	>0.05
RMSEA	0.045	<0.08
GFI	0.992	>0.9
AGFI	0.972	>0.9
NFI	0.989	≥0.95
CFI	0.994	≥0.95
IFI	0.994	≥0.95
TLI	0.985	≥0.95
SRMR	0.024	<0.05

RMSEA: Root-mean Square Error of Approximation; GFI: Goodness of Fit Index; AGFI: Adjusted Goodness of Fit Index; NFI: Normed Fit Index; CFI: Comparative Fit Index; IFI: Incremental Fit Index; TLI: Tucker-Lewis Index; SRMR: Standardized Root Mean Square Residual; χ^2^/df: chi square/degrees of freedom.

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
