# Peer review of "Mediating Effects of Emotional Symptoms on the Association between Homophobic Bullying Victimization and Problematic Internet/Smartphone Use among Gay and Bisexual Men in Taiwan"

_ijerph, 2020, doi:10.3390/ijerph17103386_

Round 1

Reviewer 1 Report

The paper "Mediating effect of emotional symptoms on the association between homophobic bullying

victimization and problematic internet / smartphone use among gay and bisexual men in Taiwan” addresses the issue of the relationship between problematic internet / smartphone use (PI / SU) among sexual minorities victims of homophobic bullying and how the outcomes are mediated by the health problem. This is an important and relevant topic because the experiences of victimization leave an indelible mark on life and in relationships. Ma the authors must enhance and update the bibliography and they must deepen the relationship betwen victimization, mental health and problematic use of the internet.

the statistical analysis model is significant and the hypothesis is confirmed.

the research plan is clearly presented but the recruitment procedures and the characteristics of the subjects should be deepened. The authors refer only to untheir previous study.

Reviewer 2 Report

Dear Authors,

I find the paper to be of high quality, with recent and valid references.

The methodology and the research conclusions presented are clear and well-established.

My only recommendations would be:

  1. when reporting on the participants' sociodemographic traits, present this data in the "Participants" section, with the "Procedure" section separately described
  2. the authors should describe whether there were any special issues with the research in the "Procedure" section of the paper
  3. when reporting correlation coefficients, please report with 2 decimal places
  4.  please report validity on all of the instruments used, as only some of the instruments used in this research have references about their previous reliability and validity (the authors have clarified this issue in the "Limitations" section of the paper, but if possible, please report previously established validity and reliability)

Respectfully

Reviewer 3 Report

Overall the article contributes important insights and discussion to the field of study but I recommend the following:

Clarify title: 

Mediating effect of emotional symptoms on the association between homophobic bullying victimization and problematic internet/smartphone use among gay and bisexual men in Taiwan

Do the authors mean

Mediating the effect of emotional symptoms on the association between homophobic bullying victimization and problematic internet/smartphone use among gay and bisexual men in Taiwan

or

Mediating effects of emotional symptoms on the association between homophobic bullying victimization and problematic internet/smartphone use among gay and bisexual men in Taiwan

Page 2:

Please give a paragraph on the context of gay, bisexual and sexual minorities in Taiwan. This background is important to avoid assumptions. 

Page 3:

"In brief, we recruited gay and bisexual men": Because the length of the paper requires brief explanations and the authors cannot go into explanatory detail, I recommend changing to "In brief, we recruited men who identified as being gay or bisexual"

"Traditional" bullying: Recommend clarifying or rewording to "Forms of bullying prior to digital media" or something similar, or give a very brief explanation of what the authors mean by "traditional bullying".

Page 9: 

"Finally, this study only recruited gay and bisexual men, so the findings may not be generalizability to other populations." Please clarify. Does 'other' refer to minority, marginalised, socially excluded populations?
